# Research on Adhesion State Sensing Method for a Gecko-Inspired Multiscale Adhesive Structure

Yanwei Liu, Bowen Li, *Member, IEEE*

*Abstract*— This study addresses the lack of real-time state perception in existing gecko-inspired adhesive structures. Inspired by the hierarchical "lamella-setae" architecture of gecko toes, a multiscale adhesive structure was developed, integrating a millimeter-scale boot-shaped elastic component with a micrometer-scale mushroom-type adhesive array. Finite element analysis based on a cohesive zone model was employed to analyze stress distribution during adhesion-detachment processes, identifying the optimal region for embedding a thin-film pressure sensor. Experimental results demonstrate that the sensing signal accurately responds to variations in normal preload and shear displacement, effectively identifying adhesion, slip, and detachment states. A gripper utilizing this technology achieved stable grasping on smooth glass, real-time state perception ("pick-and-place", "drop", "pick failure"), and controllable switching between "strong adhesion" and "easy detachment". This research provides a crucial step towards adaptive and reliable robotic manipulation in unstructured environments.

*Keywords—gecko-inspired adhesive; controllable adhesion; multiscale structure; adhesion sensing*

## I. INTRODUCTION

Geckos exhibit remarkable climbing capabilities due to their multi-level "lamella-setae" structure and precise adhesion control mechanisms, primarily relying on van der Waals forces [1]-[3]. Their ability to switch between strong adhesion and easy detachment by adjusting toe posture [4][5] provides significant inspiration for robotic grasping applications. Research on bio-inspired dry adhesives has progressed substantially, from early developments of high-aspect-ratio micropillars [6] and mushroom-shaped arrays [7] to more recent asymmetric structures such as tilted [8] and wedge-shaped arrays [9-10] for directional adhesion control. Despite these advances, most studies have focused primarily on the micro-scale, often overlooking the macro-mechanical control analogous to the function of gecko lamellae. Furthermore, while the integration of sensing materials such as conductive polymers [11] or carbon nanotubes [12] has enabled normal force perception, the synergistic sensing of both normal and shear states within a controllable, multiscale system remains challenging. This work bridges this gap by presenting a comprehensive approach—from bio-inspired design and simulation to sensor integration and experimental

validation—enabling controllable adhesion with integrated state perception for reliable robotic manipulation.

## II. BIO-INSPIRED DESIGN & FABRICATION

Inspired by the gecko toe structure (Fig. 1a), our adhesive design combines a compliant, millimeter-scale boot-shaped component for macro-scale adaptation and motion control with a dense array of micrometer-scale mushroom-shaped pillars providing high adhesion strength (Fig. 1b). This architecture enables controllable "strong adhesion" through shear motion and "easy detachment" via reverse shear.

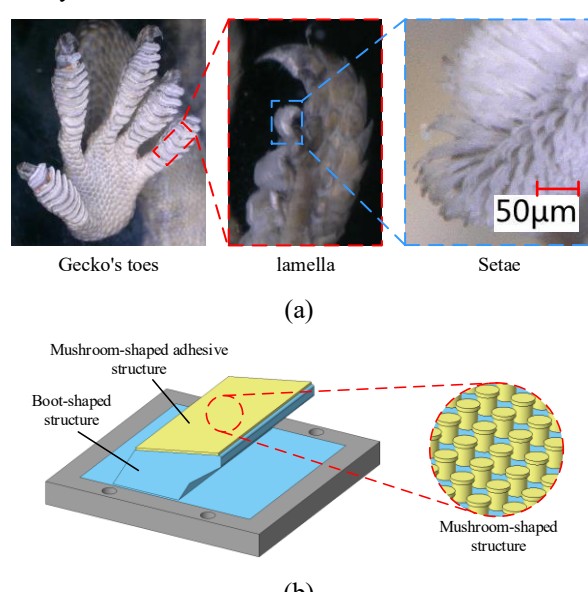

Fig. 1. (a) Gecko toe lamella-setae hierarchy. (b) Design of the bio-inspired multiscale adhesive structure.

The fabrication process incorporates a flexible thin-film piezoresistive pressure sensor within the structure (Fig. 2a). The sensor is embedded into the silicone backing layer during the casting process (Fig. 2b), enabling monitoring of internal stress states critical for adhesion perception.

The physical object of the adhesive structure with sensing capability is shown in Fig. 2(b). Due to the combined effect of internal stress generated during the curing of the silicone and the pressure from the mating surface when the adhesive structure is assembled on the test bench, the thin-film pressure sensor is initially kept under a compressed state. This allows the sensor to detect both compressive and tensile states in the region.

*Research supported by the National Natural Science Foundation of China under Grant 62403295, and the Natural Science Basic Research Program of Shaanxi Province under Grant 2023-JC-YB-339.

Y. Liu is with the School of Mechanical and Precision Instrument Engineering, Xi'an University of Technology, Xi'an, Shaanxi, 710048 China. (e-mail: liuyw@xaut.edu.cn).

B. Li is with the School of Mechanical and Precision Instrument Engineering, Xi'an University of Technology, Xi'an, Shaanxi, 710048 China. He is a Master candidate (e-mail: 1255132424@qq.com).

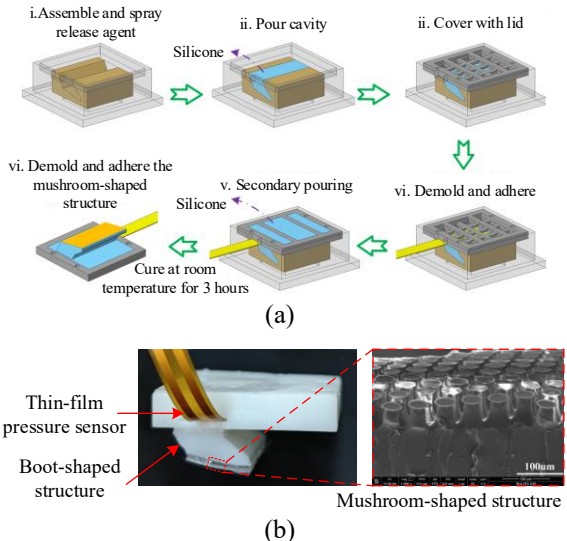

(a)

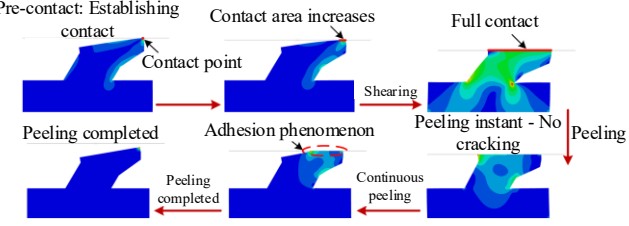

(b)

Fig. 2. (a) Fabrication schematic with integrated sensor. (b) Physical image of the sensor-integrated adhesive structure.

## III. SIMULATION ANALYSIS & SENSING STRATEGY

A finite element model (CPS4R elements, Ogden material model) incorporating a cohesive zone model was established to analyze stress distribution during adhesion and detachment cycles (Fig. 3). The simulation accurately replicates the preloading, shearing, and peeling phases, illustrating the evolution of contact area and internal stresses.

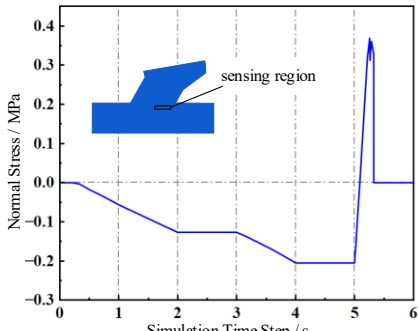

Fig. 3. Simulated adhesion and peeling processes of the adhesive structure.

Through analysis of the normal stress distribution in the backing layer, the optimal placement location for the sensor was determined. As shown in Fig.4 and Fig.5, this sensing region exhibits a distinct and characteristic transition from compressive stress (during preload and shear phases) to tensile stress (during the initiation of peeling), providing a clear and discriminative signal for state identification.

Fig. 4. Normal stress curves across different regions of the adhesive structure.

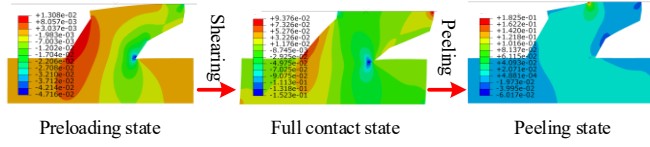

Fig. 5. Variation of normal stress in the selected sensing region during adhesion-detachment.

## IV. EXPERIMENTAL RESULTS

An experimental test platform was constructed to validate the sensing performance. The voltage signal output from the embedded sensor demonstrated strong correlation with directly measured interfacial forces.

Figure 6 illustrates the signal characteristics during a successful adhesion cycle: the signal increases proportionally with applied preload, exhibits a sharp rise during shear displacement, and shows a distinct transition during the peeling phase.

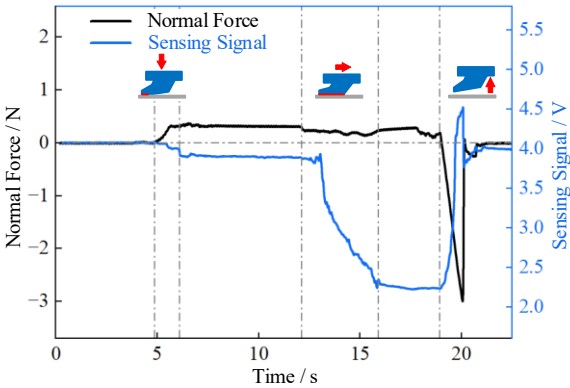

Fig. 6. Normal force and sensing signal during preloading, shearing, and peeling.

The developed structure was successfully integrated into an opposing-grip adhesive gripper. The integrated sensing system enabled real-time distinction between fundamental manipulation scenarios—"pick-and-place", "drop", and "pick failure"—based on characteristic signal signatures during object engagement and lifting phases.

## V. CONCLUSION

This study developed and validated a gecko-inspired multiscale adhesive structure with integrated state perception capability. Computational analysis using FEA identified the optimal sensor placement strategy based on stress distribution patterns. Experimental results confirmed the sensor's ability to reliably detect key states including preload magnitude, shear displacement, slip events, and detachment initiation. The technology, demonstrated through a functional gripper prototype, provides a significant advancement towards achieving adaptive and reliable robotic manipulation in unstructured environments using bio-inspired adhesive technologies.

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
