# OpenReview forum: "Research on Adhesion State Sensing Method for a Gecko-Inspired  Multiscale Adhesive Structure"
_IEEE.org/IROS/2025/Workshop/Tactile_Sensing — IROS 2025 Workshop Tactile Sensing Poster_

### Official Review · Reviewer_aiNE · 2025-09-20
**Review for Adhesion State Sensing**

**Rating:** 6
**Confidence:** 3

**Review:**

This work proposes a sensing strategy for perceiving adhesion states by incorporating pressure sensors and adhesive microstructures. The motivation is interesting, while there are some issues as follows.
1. The thin-film sensor is embedded during the casting process. How can it detect negative normal forces in the peeling process?
2. How to solve the normal forces based on the obtained sensing signals? Is there an explicit expression?
3. Please provide more details on how to conduct the successful adhesion and failed adhesion in Fig. 6.

---

### Official Review · Reviewer_EUW8 · 2025-09-25
**Sensor calibration needed for normal and shear loads**

**Rating:** 6
**Confidence:** 4

**Review:**

This paper presents an interesting bio-inspired adhesive structure with integrated tactile sensing, which is important for state estimation/monitoring for soft robots. The preliminary results show that the sensors can respond to loadings along different directions. However, it is unclear what is the sensing mechanism to differentiate normal and shear loads, and how the sensor will respond when there is both normal and shear loads. The work would be strengthened by a thorough sensor calibration under distinct and combined loading conditions to address this ambiguity.

---

### Official Review · Reviewer_WPFP · 2025-09-25
**Review for Adhesion State Sensing Method**

**Rating:** 9
**Confidence:** 5

**Review:**

This work proosed one kind of adhesion state sensing method for a Gecko-Inspired multiscale adhesive structure. Both simulation and experimental results have demonstrated the effectiveness of the method. The research is very interesting.